# ManiSkill: Generalizable Manipulation Skill Benchmark with Large-Scale Demonstrations

**Tongzhou Mu**[*], **Zhan Ling**[*], **Fanbo Xiang**[*], **Derek Yang**[*], **Xuanlin Li**[*],
**Stone Tao, Zhiao Huang, Zhiwei Jia, Hao Su**
University of California San Diego
{t3mu, z6ling, fxiang, d7yang, xul012, stao, z2huang, zjia, haosu}@ucsd.edu

https://github.com/haosulab/ManiSkill

## Abstract

Object manipulation from 3D visual inputs poses many challenges on building generalizable perception and policy models. However, 3D assets in existing benchmarks mostly lack the diversity of 3D shapes that align with real-world intra-class complexity in topology and geometry. Here we propose SAPIEN **Mani**pulation **Skill** Benchmark (ManiSkill) to benchmark manipulation skills over diverse objects in a full-physics simulator. 3D assets in ManiSkill include large intra-class topological and geometric variations. Tasks are carefully chosen to cover distinct types of manipulation challenges. Latest progress in 3D vision also makes us believe that we should customize the benchmark so that the challenge is inviting to researchers working on 3D deep learning. To this end, we simulate a moving panoramic camera that returns ego-centric point clouds or RGB-D images. In addition, we would like ManiSkill to serve a broad set of researchers interested in manipulation research. Besides supporting the learning of policies from interactions, we also support learning-from-demonstrations (LfD) methods, by providing a large number of high-quality demonstrations (~36,000 successful trajectories, ~1.5M point cloud/RGB-D frames in total). We provide baselines using 3D deep learning and LfD algorithms. All code of our benchmark (simulator, environment, SDK, and baselines) is open-sourced, and a challenge facing interdisciplinary researchers will be held based on the benchmark.

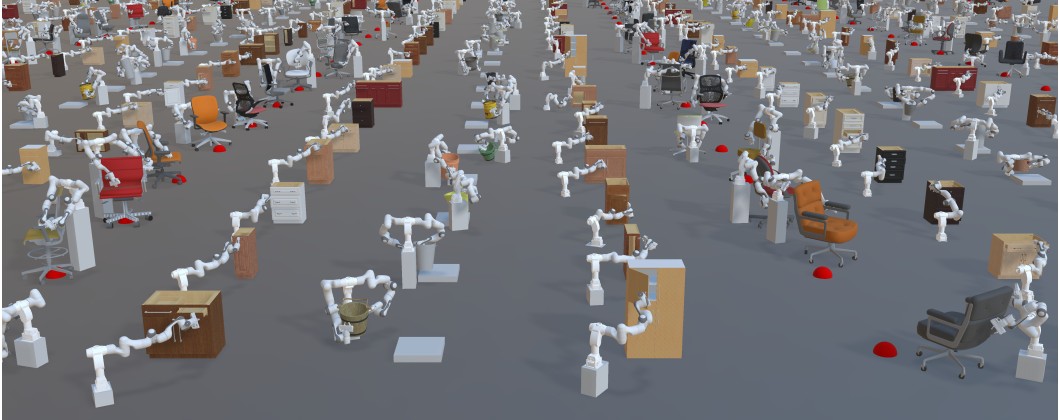

Figure 1: A subset of environments in ManiSkill. We currently support 4 different manipulation tasks: OpenCabinetDoor, OpenCabinetDrawer, PushChair, and MoveBucket; each features a large variety of 3D articulated objects to encourage generalizable physical manipulation skill learning.

---

[*]These authors contributed equally to this work

35th Conference on Neural Information Processing Systems (NeurIPS 2021) Track on Datasets and Benchmarks.

# 1 Introduction and Related Works

To automate repetitive works and daily chores, robots need to possess human-like manipulation skills. A remarkable feature of human manipulation skill is that, once we have learned to manipulate a category of objects, we will be able to manipulate even unseen objects of the same category, despite the large topological and geometric variations. Taking swivel chairs as an example, regardless of the existence of armrest or headrest, the number of wheels, or the shape of backrest, we are confident of using them immediately. We refer to such ability to interact with unseen objects within a certain category as **generalizable manipulation skills**.

Generalizable manipulation skill learning is at the nexus of vision, learning, and robotics, and poses many interesting research problems. Recently, this field has started to attract much attention across disciplines. For example, reinforcement learning and imitation learning are applied to object grasping and manipulation [30, 61, 35, 56, 34, 5, 38, 49, 75]. On the other hand, [42, 54, 47, 18, 40, 63, 55, 19] can propose novel grasp poses on novel objects based on visual inputs. To further foster synergistic efforts, it is crucial to build a benchmark that backs reproducible research and allows researchers to compare and thoroughly examine different algorithms.

However, building such a benchmark is extremely challenging. To motivate our benchmark proposal, we first analyze key factors that complicate the design of generalizable manipulation skill benchmarks and explain why existing benchmarks are still insufficient. With the motivations in mind, we then introduce design features of our SAPIEN Manipulation Skills Benchmark (abbreviated as **ManiSkill**).

**Key factors that affect our benchmark design.** To guide users and create concentration on algorithm design, four key factors must be considered: 1) manipulation policy structure, 2) diversity of objects and tasks, 3) targeted perception algorithms, and 4) targeted policy algorithms.

1) Manipulation policy structure: Manipulation policies have complex structures that require different levels of simulation support, and we focus on full-physics simulation. Since simulating low-level physics is difficult, many robot simulators only support abstract action space (i.e., manipulation skills already assumed) [32, 69, 50, 71, 70, 59, 24, 17, 69, 24]. It is convenient to study high-level planning in these benchmarks; however, it becomes impossible to study more challenging scenarios with high-dimensional and complex low-level physics. Some recent benchmarks [7, 62, 79, 66, 74, 28] start to leverage the latest full-physics simulators [65, 14, 3, 57] to support physical manipulation. Despite the quantity of existing environments, most of them lack the ability to benchmark object-level generalizability within categories, and lack inclusion for different methodologies in the community, while we excel in these dimensions, which is explained next.

2) Diversity of objects and tasks: To test object-level generalizability, the benchmark must possess enough intra-class variation of object topology, geometry, and appearance, and we provide such variation. Several benchmarks or environments, including robosuite [79], RLBench [28], and Meta-World [74], feature a wide range of tasks; however, they possess a common problem: lacking object-level variations. Among past works, DoorGym [66] is equipped with the best object-level variations: it is a door opening benchmark with doors procedurally generated from different knob shapes, board sizes, and physical parameters, but it still does not capture some simple real-world variations, such as multiple doors with multiple sizes on cabinets with different shapes. This is in part due to the limitations of procedural modeling. Even though procedural modeling has been used in 3D deep learning [16, 73], it often fails to cover objects with real-world complexity, where crowd-sourced data from Internet users and real-world scans are often preferred (which is our case). Finally, a single type of task like opening doors cannot cover various motion types. For example, pushing swivel chairs requires very different skills from opening doors since it involves controlling under-actuated systems through dual-arm collaboration. Therefore, it is essential to build benchmarks with *both* great asset variations and wide skill coverage.

3) Targeted perception algorithms: Benchmarks need to decide the type and format of sensor data, and we focus on 3D sensor data mounted on robots. Many existing benchmarks, such as *DoorGym*, rely on fixed cameras to capture 2D images; however, this setting greatly limits the tasks a robot can solve. Instead, robot-mounted cameras are common in the real world to allow much higher flexibility, such as Kinova MOVO [2], and autonomous driving in general; those cameras are usually designed to capture 3D inputs, especially point clouds. Moreover, tremendous progress has been achieved to build neural networks with 3D input [52, 53, 64, 25, 12, 77, 51], and these 3D networks have demonstrated strong performance (e.g., they give better performance than 2D image networks

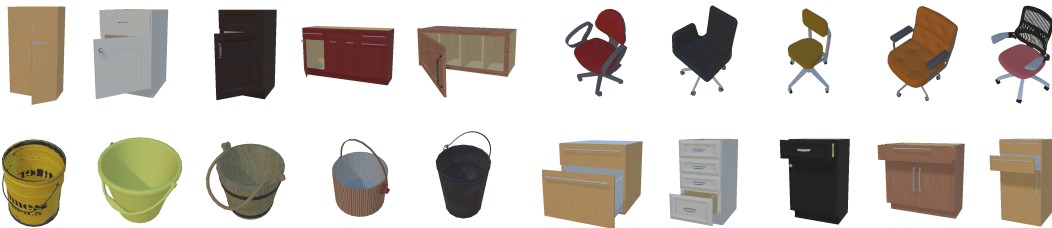

Figure 2: ManiSkill features diverse articulated objects with complex topological and geometric variations, such as different numbers and shapes of doors and/or drawers on different shapes of cabinets. We invested significant effort to process objects from the PartNet-Mobility Dataset and integrate into our tasks, such as adjusting the size and physical parameters (e.g. friction) so that environments are solvable, along with manual convex decomposition.

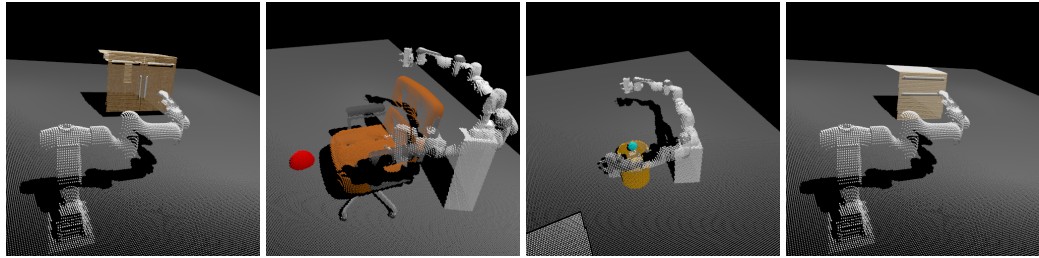

Figure 3: Rendered point clouds from our tasks. ManiSkill supports 3D visual inputs which are widely accessible in real environments, allowing various computer vision models to be applied. (For better view, we show point clouds obtained from cameras mounted in the world frame. In actual tasks, cameras are mounted on the robot head, offering an egocentric view.)

on autonomous driving datasets [68]). [11, 8, 48, 39, 44, 76] have also adopted 3D deep learning models for perceiving and identifying kinematic structures and object poses for articulated object manipulation. Our benchmark provides users with an ego-centric panoramic camera to capture point cloud / RGB-D inputs. Additionally, we present and evaluate 3D neural network-based policy learning baselines.

4) Targeted policy algorithms: Different policy learning algorithms require different training data and settings, and we provide multiple tracks to advocate for fair comparison. For example, imitation learning [27, 58, 56] and offline RL [23, 33, 36] can learn a policy purely from demonstrations datasets [15, 9], but online RL algorithms [26, 60] require interactions with environments. Therefore, a clear and meaningful split of tracks can encourage researchers with different backgrounds to explore generalizable manipulation skills and let them focus on different aspects of the challenge, e.g, network design, perception, interaction, planning, and control. While oftentimes other benchmarks are limited to a single domain of research and a single modality, our benchmark supports three different tracks for researchers from computer vision, reinforcement learning, and robotics fields.

**Our benchmark.** Above we discussed factors of manipulation benchmark design and mentioned the principles behind ManiSkill. Here we introduce the key features of the benchmark. ManiSkill is a large-scale open-source benchmark for physical manipulation skill learning over a diverse set of articulated objects from 3D visual inputs. ManiSkill has four main features: **First**, to support generalizable policy learning, ManiSkill provides objects of high topology and geometry variations, as shown in Fig 2. It currently includes a total of 162 objects from 3 object categories (more objects are being added) selected and manually processed from a widely used 3D vision dataset. **Second**, ManiSkill focuses on 4 object-centric manipulation tasks that exemplify household manipulation skills with different types of object motions, thereby posing challenges to distinct aspects of policy design/learning (illustrated in Fig 1 and Fig 3). As an ongoing effort, we are designing more. **Third**, to facilitate learning-from-demonstration methods, we have collected a large number of successful trajectories (~36,000 trajectories, ~1.5M 3D point cloud / RGB-D frames in total). **Fourth**, the environments feature high-quality data for physical manipulation. We take significant efforts to select, fix, and re-model the original PartNet-Mobility data [72, 46, 10], as well as design the reward generation rules, so that the manipulation task of each object can be solved by an RL algorithm.

A major challenge to build our benchmark is to collect demonstrations. Some tasks are tricky (e.g., swivel chair pushing requires dual-arm coordination), and it is difficult and unscalable to manually control the robots to collect large-scale demos. It is also unclear whether traditional motion planning pipelines can solve all tasks. Thankfully, reinforcement learning does work for individual objects, allowing for a divide-and-conquer approach to create high-quality demonstrations. With a meticulous effort on designing a shared reward template to automatically generate reward functions for all object instances of each task and executing an RL agent for each object instance, we are able to collect a large number of successful trajectories. This RL plus divide-and-conquer approach is very scalable with respect to the number of object instances within a task, and we leave cross-task RL reward design for future work. It is worth noting that, *we do NOT target at providing a GENERIC learning-from-demonstrations benchmark* that compares methods from all dimensions. Instead, we compare the ability of different algorithms to utilize our demonstrations to solve manipulation tasks.

Another important feature of ManiSkill is that it is completely free and built on an entirely open-source stack. Other common **physical** manipulation environments, including robosuite[79], DoorGym[66], MetaWorld[74], and RLBench[28], depend on commercial software.

To summarize, here are the key contributions of ManiSkill Benchmark.

- The topology and geometry variation of our data allow our benchmark to compare object-level generalizability of different physical manipulation algorithms. Our data is high-quality, that every object is verified to support RL.

- The manipulation tasks we design target at distinct challenges of manipulation skills (by motion types, e.g. revolute and prismatic joint constraints, or by skill properties, e.g. requirement of dual-arm collaboration).

- ManiSkill provides a large-scale demonstrations dataset with ~36,000 trajectories and ~1.5M point cloud/RGB-D frames to facilitate learning-from-demonstrations approaches. The demonstrations are collected by a scalable RL approach with dense rewards generated by a shared reward template within each task.

- We provide several 3D deep learning-based policy baselines.

## 2 ManiSkill Benchmark

The goal of building ManiSkill benchmark can be best described as facilitating *learning generalizable manipulation skills from 3D visual inputs with demonstrations*. "*Manipulation*" involves low-level physical interactions and dynamics simulation between robot agents and objects; "*skills*" refer to short-horizon physics-rich manipulation tasks, which can be viewed as basic building blocks of more complicated policies; "*3D visual inputs*" are egocentric point cloud and / or RGB-D observations captured by a panoramic camera mounted on a robot; "*demonstrations*" are trajectories that solve tasks successfully to facilitate learning-from-demonstrations approaches.

In this section, we will describe the components of ManiSkill benchmark in detail, including basic terminologies and setup, task design, demonstration trajectory collection, training-evaluation protocol, and asset postprocessing with verification.

| Task | # objects | | | Dual-arm Collaboration? | Solvable by Motion Planning? | DoF* |
|---|---|---|---|---|---|---|
| | All | Train | Test | | | |
| OpenCabinetDoor | 52(82) | 42(66) | 10(16) | No | Easy | 1 |
| OpenCabinetDrawer | 35(70) | 25(49) | 10(21) | No | Easy | 1 |
| PushChair | 36 | 26 | 10 | Yes | Hard | ~15-25 |
| MoveBucket | 39 | 29 | 10 | Yes | Medium | 7 |

Table 1: Dataset statistics for ManiSkill. For OpenCabinetDoor and OpenCabinetDrawer, numbers outside of the parenthesis indicate the number of unique cabinets, where each cabinet may have more than one door/drawer. Numbers in the parenthesis indicate the total number of doors/drawers. * The DoF in the table indicates the DoF involved in solving a task. For OpenCabinetDoor and OpenCabinetDrawer, an agent only needs to open one designated door/drawer. For PushChair and MoveBucket, 6 extra DoF are included since chairs and buckets can move freely in 3D space.

## 2.1 Basic Terminologies and Setup

In ManiSkill, a **task** or a **skill** $\mathcal{T} = \{T_{o,l} : o \in \mathcal{O}, l \in \mathcal{L}_o\}$ consists of finite-horizon POMDPs (Partially Observable Markov Decision Processes) defined over a set of objects $\mathcal{O}$ of the same category (e.g., chairs) and a set of environment parameters $\mathcal{L}_o$ associated with an object $o \in \mathcal{O}$ (e.g. friction coefficients of joints on a chair). An **environment** is a set of POMDPs $\mathcal{E}_o = \{T_{o,l} : l \in \mathcal{L}_o\}$ defined over a single object $o$ and its corresponding parameters. Each $T_{o,l}$ is a specific instance of an environment, represented by a tuple of sets $(S, A, P, R, O)$. Here, $s \in S$ is an environment state that consists of *robot states* (e.g. joint angles of the robot) and *object states* (e.g. object pose and the joint angles); $a \in A$ is an action that can be applied to a robot (e.g. target joint velocity of a velocity controller); $P(s'|s,a)$ is the physical dynamics; $R$ is a binary variable that indicates if the task is successfully solved; $O(o|s)$ is a function which generates observations from an environment state, and it supports three modes in ManiSkill: state, pointcloud, and rgbd. In state mode, the observation is identical to $s$. In pointcloud and rgbd modes, the *object states* in $s$ are replaced by the corresponding point cloud / RGB-D visual observations captured from a panoramic camera mounted on a robot. state mode is not suitable for studying generalizability, as *object states* are not available in realistic setups, where information such as object pose has to be estimated based on some forms of visual inputs that are universally obtainable (e.g. point clouds and RGB-D images).

For each task, objects are partitioned into training objects $\mathcal{O}_{train}$ and test objects $\mathcal{O}_{test}$, and environments are divided into training environments $\{T_{o,l}|o \in \mathcal{O}_{train}\}$ and test environments $\{T_{o,l}|o \in \mathcal{O}_{test}\}$. For each training environment, successful demonstration trajectories are provided to facilitate learning-from-demonstrations approaches.

We define **object-level generalizable manipulation skill** as a manipulation skill that can generalize to unseen test objects after learning on training objects where the training and test objects are from the same category. Some notable challenges of our tasks come from partial observations (i.e. point clouds / RGB-D images only covering a portion of an object), robot arms occluding parts of an object, and complex shape understanding over objects with diverse topological and geometric properties.

## 2.2 Tasks with Diverse Motions and Skills

Object manipulation skills are usually associated with certain types of desired motions of target objects, e.g, rotation around an axis. Thus, the tasks in ManiSkill are designed to cover different types of object motions. We choose four common types of motion constraints: revolute joint constraint, prismatic joint constraint, planar motion constraint, and no constraints, and build four tasks to exemplify each of these motion types. In addition, different tasks also feature different properties of manipulation, such as dual-arm collaboration and solvability by motion planning. Statistics for our tasks are summarized in Table 1. Descriptions for our tasks are stated below (more details in Sec B of supplementary).

**OpenCabinetDoor** exemplifies motions constrained by a revolute joint. In this task, a single-arm robot is required to open a designated door on a cabinet. The door motion is constrained by a revolute joint attached to the cabinet body. This task is relatively easy to solve by traditional motion planning and control pipelines, so it is suitable for comparison between learning-based methods and motion planning-based methods.

**OpenCabinetDrawer** exemplifies motions constrained by a prismatic joint. This task is similar to OpenCabinetDoor, but the robot needs to open a target drawer on a cabinet. The drawer motion is constrained by a prismatic joint attached to the cabinet body.

**PushChair** exemplifies motions constrained on a plane through wheel-ground contact. A dual-arm robot needs to push a swivel chair to a target location on the ground and prevent it from falling over. PushChair exemplifies the ability to manipulate complex underactuated systems, as swivel chairs generally have many joints, resulting in complex dynamics. Therefore, it is difficult to solve PushChair by motion planning and favors learning-based methods.

**MoveBucket** exemplifies motions without constraints. In this task, a dual-arm robot needs to move a bucket with a ball inside and lift it onto a platform. There are no constraints on the motions of the bucket. However, this task is still very challenging because: 1) it heavily relies on two-arm coordination as the robot needs to lift the bucket; 2) the center of mass of the bucket-ball system is consistently changing, making balancing difficult.

Note that all environments in ManiSkill are verified to be solvable, i.e., for each object, we guarantee that there is a way to manipulate it to solve the corresponding task. This is done by generating successful trajectories in each environment (details in Sec 2.4). Instead of creating lots of tasks but leaving the solvability problems to users, our tasks are constructed with appropriate difficulty and verified solvability.

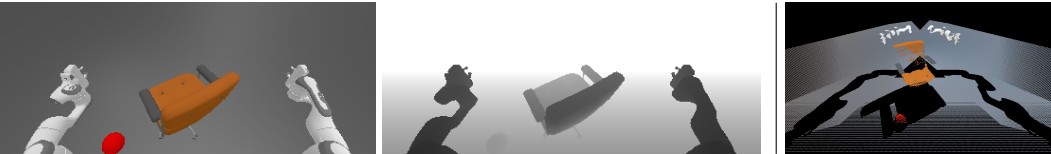

Figure 4: RGB-D (RGB/Depth) and point cloud observations in ManiSkill. Left two images: RGB-D image from *one* of the three cameras mounted on the robot. The three cameras together provide an ego-centric panoramic view. Right image: visualization of fused point cloud from *all* three cameras. The center of robot body cannot be seen since the captured point cloud comes from an ego-centric view. Parts of the chair are occluded by itself (as cameras are mounted on the robot).

## 2.3 Robots, Actions, Visual Observations, and Rewards

All the tasks in ManiSkill use similar robots, which are composed of three parts: moving platform, Sciurus [4] robot body, and one or two Franka Panda [1] arm(s). The moving platform can move and rotate on the ground plane, and its height is adjustable. The robot body is fixed on top of the platform, providing support for the arms. Depending on the task, one or two robot arm(s) are connected to the robot body. There are 22 joints in a dual-arm robot and 13 for a single-arm robot. To match realistic robotics setups, we use PID controllers to control the joints of robots. The action space corresponds to the normalized target values of all controllers. In addition to joint space control, ManiSkill supports operational space control, i.e., directly controlling the end-effectors in the Cartesian space.

As mentioned in Sec 2.1, ManiSkill supports three observation modes: `state`, `pointcloud`, and `rgbd`, where the latter two modes are suitable for studying object-level generalizability. The RGB-D and point cloud observations are captured from three cameras mounted on the robot to provide an ego-centric panoramic view, resembling common real-world robotics setups. The three cameras are 120° apart from each other, and the resolution of each camera is 400×160. The observations from all cameras are combined to form a final panoramic observation. Visualizations of RGB-D / point cloud observations are shown in Fig 4. In addition, we provide some task-relevant segmentation masks for both RGB-D and point cloud observations (details in Sec B.3 of supplementary).

ManiSkill supports two kinds of rewards: `sparse` and `dense`. A sparse reward is a binary signal which is equivalent to a task-specific success condition. Learning with sparse rewards is very difficult. To alleviate such difficulty, we carefully designed well-shaped dense reward functions for each task (details in Sec C.1 of supplementary). The dense rewards are also used in demonstration collection, which will be elaborated in Sec 2.4.

## 2.4 RL-Based Demo Collection and MPC-Assisted Reward Template Design

Our interactive environment naturally supports methods such as reinforcement learning or classical robotics pipelines. However, to build an algorithm with object-level generalizability either by RL or by designing rules, it is probably prohibitively complex and resource-demanding for many researchers interested in manipulation learning research (e.g., vision researchers might be primarily interested in the perception module). We observe that many learning-from-demonstrations algorithms (e.g., behavior cloning) are much easier to start with and require less resources.

To serve more researchers, ManiSkill provides a public demonstration dataset with a total of ~36,000 successful trajectories and ~1.5M frames (300 trajectories for each training object in each task). The demonstrations are provided in the format of internal environment states to save storage space, and users can render the corresponding point cloud / RGB-D frames using the provided scripts.

In order to construct this large-scale demonstration dataset, we need a **scalable** pipeline that can produce plentiful demonstrations automatically. Compared to other existing approaches (e.g. human annotation by teleoperation, motion planning), we adopted a reinforcement learning (RL)-based

pipeline, which requires significantly less human effort and can generate an arbitrary number of demonstrations at scale. Though it is common to collect demonstrations by RL [20, 41, 37, 27, 6, 31, 29, 80], directly training a single RL agent to collect demonstrations for all environments in ManiSkill is challenging because of the large number of different objects and the difficulty of our manipulation tasks. To collect demonstrations at scale, we design an effective pipeline as follows.

Our pipeline contains two stages. In the first stage, we need to design and verify dense rewards for RL agents. For each task, we design a shared reward template based on the skill definitions of this task with human prior. *Note that this reward template is shared across all the environments (objects) in a task instead of manually designed for each object.* In order to *quickly verify* the reward template (as our tasks are complicated and solving by RL takes hours), we use Model-Predictive Control (MPC) via Cross Entropy Method (CEM), which can be efficiently parallelized to find a trajectory within 15 minutes (if successful) from *one single initial state* using 20 CPUs. While MPC is an efficient tool to verify our reward template, it is not suitable for generating our demonstrations dataset, which should contain diverse and randomized initial states. This is because MPC has to be retrained independently each time to find a trajectory from each of the 300 initial states for each training object of each task, rendering it unscalable. Therefore, in the second stage, we train model-free RL agents to collect demonstrations. We also found that training one single RL agent on many environments (objects) of a task is very challenging, but training an agent to solve a single specific environment is feasible and well-studied. Therefore, we collect demonstrations in a divide-and-conquer way: for each environment, we train an SAC [26] agent and generate successful demonstrations. More details can be found in Sec C of supplementary.

## 2.5 Multi-Track Training-Evaluation Protocol

As described in Sec 2.1, agents are trained on the training environments with their corresponding demonstrations and evaluated on the test environments for object-level generalizability, under `pointcloud` or `rgbd` mode. Moreover, ManiSkill benchmark aims to encourage interdisciplinary insights from computer vision, reinforcement learning, and robotics to advance generalizable physical object manipulation. To this end, we have developed our benchmark with 3 different tracks:

**No Interactions**: this track requires solutions to only use our provided demonstration trajectories during training. No interactions (i.e. additional trajectory collection, online training, etc.) are allowed. For this track, solutions may choose to adopt a simple but effective supervised learning algorithm — matching the predicted action with the demonstration action given visual observation (i.e. behavior cloning). Therefore, this track encourages researchers to explore 3D computer vision network architectures for generalizable shape understanding over complex topologies and geometries.

**No External Annotations**: this track allows online model fine-tuning over training environments on top of No Interactions track. However, the solution must not contain new annotations (e.g. new articulated objects from other datasets). This track encourages researchers to explore online training algorithms, such as reinforcement learning with online data collection.

**No Restrictions**: this track allows solutions to adopt any approach during training, such as labelling new data and creating new environments. Researchers are also allowed to use manually designed control and motion planning rules, along with other approaches from traditional robotics.

The benchmark evaluation metric is the *mean success rate* on a predetermined set of test environment instances. For each task, we have defined the success condition (described in Sec B.4 of supplementary), which is automatically reported in the evaluation script provided by us. Each track should be benchmarked separately.

## 2.6 Asset Selection, Re-Modeling, Postprocessing, and Verification

While the PartNet-Mobility dataset (from SAPIEN [72]) provides a repository of articulated object models, the original dataset can only provide full support for vision tasks such as joint pose estimation. Therefore, we take significant efforts to select, fix, and verify the models.

First, PartNet-Mobility dataset is not free of annotation errors. For example, some door shafts are annotated at the same side as the door handles. While it does not affect the simulation, such models are unnatural and not good candidates to test policy generalizability. Thus, we first render all assets and manually exclude the ones with annotation errors.

Moreover, fast simulation requires convex decomposition of 3D assets. However, the automatic algorithm used in the original SAPIEN paper, VHACD [43], cannot handle all cases well. For example, VHACD can introduce unexpected artifacts, such as dents on smooth surfaces, which agents can take advantage of. To fix the errors, we identify problematic models by inspection and use Blender's [13] shape editing function to manually decompose the objects.

Even with all the efforts above, some models can still present unexpected behaviors. For example, certain cabinet drawers may be stuck due to inaccurate overlapping between collision shapes. Therefore, we also verify each object by putting them in the simulator and learn a policy following Sec 2.4. We fix issues if we cannot learn a policy to achieve the task. We repeat until all models can yield a successful policy by MPC.

# 3 Baseline Architectures, Algorithms, and Experiments

Learning object-level generalizable manipulation skills through 3D visual inputs and learning-from-demonstrations algorithms has been underexplored. Therefore, we designed several baselines and open-sourced their implementations here to encourage future explorations in the field.

We adopted `pointcloud` observation mode and designed point cloud-based vision architectures as our feature extractor since previous work [68] has achieved significant performance improvements by using point clouds instead of RGB-D images. Point cloud features include position, RGB, and segmentation masks (for the details of segmentation masks, see Sec B.3 in the supplementary), and we concatenate the *robot state* to the features of each point. Intuitively, this allows the extracted feature to not only contain geometric information of objects, but also contain the relation between the robot and each individual object, such as the closest point to the robot, which is very difficult to be learned without such concatenation. In addition, we downsample the point cloud data to increase training speed and reduce the memory footprint (see Sec D.1 of supplementary).

The first point cloud-based architecture uses one single PointNet [52], a very popular 3D deep learning backbone, to extract a global feature for the entire point cloud, which is fed into the final MLP. The second architecture uses different PointNets to process points belonging to different segmentation masks. The global features from the PointNets are then fed into a Transformer [67], after which a final attention pooling layer extracts the final representations and feeds into the final MLP. We designed and benchmarked this architecture since it allows the model to capture the relation between different objects and possibly provides better performance. Details of the architectures are presented in Sec D.2 of the supplementary material, and a detailed architecture diagram of PointNet + Transformer is presented in Fig 7 of the supplementary material. While there is a great room to improve, we believe that these architectures could provide good starting points for many researchers.

For learning-from-demonstrations algorithms on top of point cloud architectures, we benchmark two approaches - Imitation Learning (IL) and Offline/Batch Reinforcement Learning (Offline/Batch RL). For imitation learning, we choose a simple and widely-adopted algorithm: behavior cloning (BC) - directly matching predicted and ground truth actions through minimizing $L_2$ distance. For offline RL, we benchmark Batch-Constrained Q-Learning (BCQ) [23] and Twin-Delayed DDPG[22] with Behavior Cloning (TD3+BC) [21]. We follow their original implementations and tune the hyperparameters. Details of the algorithm implementations are presented in Sec D of the supplementary material.

## 3.1 Single Environment Results

| #Demo Trajectories | 10 | 30 | 100 | 300 | 1000 |
|---|---|---|---|---|---|
| #Gradient Steps | 2000 | 4000 | 10000 | 20000 | 40000 |
| PointNet, BC | 0.13 | 0.23 | 0.37 | 0.68 | 0.76 |
| PointNet + Transformer, BC | 0.16 | 0.35 | 0.51 | 0.85 | 0.90 |
| PointNet + Transformer, BCQ | 0.02 | 0.05 | 0.23 | 0.45 | 0.55 |
| PointNet + Transformer, TD3+BC | 0.03 | 0.13 | 0.22 | 0.31 | 0.57 |

Table 2: The average success rates of different agents on **one single environment** (fixed object instance) of OpenCabinetDrawer with different numbers of demonstration trajectories. The average success rates are calculated over 100 evaluation trajectories. While network architectures and algorithms play an important role in the performance, learning manipulation skills from demonstrations is challenging without a large number of trajectories, even in one single environment.

As a glimpse into the difficulty of learning manipulation skills from demonstrations in our benchmark, we first present results with an increasing number of demonstration trajectories on **one single environment** of OpenCabinetDrawer in Table 2. We observe that the success rate gradually increases as the number of demonstration trajectories increases, which shows the agents can indeed benefit from more demonstrations. We also observe that inductive bias in network architecture plays an important role in the performance, as PointNet + Transformer is more sample efficient than PointNet. Interestingly, we did not find offline RL algorithms to outperform BC. We conjecture that this is because the provided demonstrations are all successful ones, so an agent is able to learn a good policy through BC. In addition, our robot's high degree of freedom and the difficulty of the task itself pose a challenge to offline RL algorithms. Further discussions on this observation are presented in Sec D.3 of the supplementary material. It is worth noting that our experiment results should not discourage benchmark users to include failure trajectories and find better usage of offline RL methods, especially those interested in the No External Annotations track described in Sec 2.5.

## 3.2 Object-Level Generalization Results

| Algorithm | BC | | | | BCQ | | TD3+BC | |
|---|---|---|---|---|---|---|---|---|
| Architecture | PointNet | | PointNet + Transformer | | PointNet + Transformer | | PointNet + Transformer | |
| Split | Training | Test | Training | Test | Training | Test | Training | Test |
| OpenCabinetDoor | 0.18±0.02 | 0.04±0.03 | 0.30±0.06 | 0.11±0.02 | 0.16±0.02 | 0.04±0.02 | 0.13±0.03 | 0.04±0.02 |
| OpenCabinetDrawer | 0.24±0.03 | 0.11±0.03 | 0.37±0.06 | 0.12±0.02 | 0.22±0.04 | 0.11±0.03 | 0.18±0.02 | 0.10±0.02 |
| PushChair | 0.11±0.02 | 0.09±0.02 | 0.18±0.02 | 0.08±0.01 | 0.11±0.01 | 0.08±0.01 | 0.12±0.02 | 0.08±0.01 |
| MoveBucket | 0.03±0.01 | 0.02±0.01 | 0.15±0.01 | 0.08±0.01 | 0.08±0.01 | 0.06±0.01 | 0.05±0.01 | 0.03±0.01 |

Table 3: Mean and standard deviation of *average success rates* on **training and test environments** of each task over 5 different runs, under the point cloud observation. Models are trained with our demonstrations dataset, with 300 demonstration trajectories per training environment. For each task, the average test success rates are calculated over the 10 test environments and 50 evaluation trajectories per environment. Obtaining one single agent capable of learning manipulation skills across multiple objects and generalizing the learned skills to novel objects is challenging.

We now present results on **object-level generalization**. We train each model for 150k gradient steps. This takes about 5 hours for BC, 35 hours for BCQ, and 9 hours for TD3+BC using the PointNet + Transformer architecture on one NVIDIA RTX 2080Ti GPU. As shown in Table 3, even with our best agent (BC PointNet + Transformer), the overall success rates on both training and test environments are low. We also observe that the training accuracy over object variations is significantly lower than the training accuracy on one single environment (in Table 2). The results suggest that existing works on 3D deep learning and learning-from-demonstrations algorithms might have been insufficient yet to achieve good performance when trained for physical manipulation skills over diverse object geometries and tested for object-level generalization. Therefore, we believe there is a large space to improve, and our benchmark poses interesting and challenging problems for the community.

## 4 Conclusion and Limitations

In this work, we propose ManiSkill, an articulated benchmark for generalizable physical object manipulation from 3D visual inputs with diverse object geometries and large-scale demonstrations. We expect ManiSkill would encourage the community to look into object-level generalizability of manipulation skills, specifically by combining cutting-edge research of 3D computer vision, reinforcement learning, and robotics.

Our benchmark is limited in the following aspects: 1) Currently, we provide 162 articulated objects in total. We plan to process more objects from the PartNet-Mobility dataset [72] and add them to our ManiSkill assets; 2) While the four tasks currently provided in ManiSkill exemplify distinct manipulation challenges, they do not comprehensively cover manipulation skills in household environments. We plan to add more tasks among the same skill properties (e.g, pouring water from one bucket to another bucket through two-arm coordination); 3) We have not conducted sim-to-real experiments yet, and this will be a future direction of ManiSkill.

## Acknowledgement

We thank Qualcomm for sponsoring the associated challenge, Sergey Levine and Ashvin Nair for insightful discussions during the whole development process, Yuzhe Qin for the suggestions on building robots, Jiayuan Gu for providing technical support on SAPIEN, and Rui Chen, Songfang Han, Wei Jiang for testing our system.

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
