# OpenReview forum: "ManiSkill: Generalizable Manipulation Skill Benchmark with Large-Scale Demonstrations"
_NeurIPS.cc/2021/Track/Datasets_and_Benchmarks/Round2 — NeurIPS 2021 Datasets and Benchmarks Track (Round 2)_

### Official Review · Reviewer_6HLA · 2021-09-19
**The benchmark provides a wealth of data and demonstrations that benefit the research community**

**Rating:** 7
**Confidence:** 3
**Correctness:** The construction of the dataset and e…
**Clarity:** The paper is well written and easy to…

**Strengths:**

* The dataset contains various geometry and topology in high quality, which is important for evaluating the generalizability of the physical manipulation algorithms.

* Challenges and demonstrations are provided in a large amount of objects and trajectories.

* The 3D deep learning-based policy baselines are presented.

**Weaknesses:**

The amount of objects categories are limited, and the four tasks are relatively simple. However, these data are sufficient as a novel benchmark for physical manipulation skill, and has the potential to be extended.

**Additional Feedback:**

None.

**Documentation:**

Details about the dataset and benchmark have been released in the project page.  There are sufficient details to support reproducibility.

**Ethics:**

No ethics problem needs to be concerned.

**Relation To Prior Work:**

The discussion about prior works are clear.

**Summary And Contributions:**

The authors propose ManiSkill to benchmark manipulation skills over diverse objects in a full-physics simulator.  Panoramic cameras are simulated that captures ego-centric point clouds or RGB-D images. The benchmark is designed to serve a broad set of researchers interested in manipulation research, and supports the learning of policies from interactions. Besides, a large number of high-quality demonstrations are provided to support learning-from-demonstrations methods.

---

> ### Author Response · Authors · 2021-09-26
> **Thanks for your encouraging feedback and suggestions!**
>
> We would like to sincerely thank you for reviewing our paper and providing constructive comments!
>
> Here we address your comments below:
>
> > The amount of objects categories are limited
>
> The object dataset construction is an ongoing effort. We are continuously processing and modeling new objects for each existing category, and we also plan to increase the number of categories of tasks in the next version of the ManiSkill benchmark.
>
> > The four tasks are relatively simple
>
> For the difficulty of our tasks, we would like to note that, as shown in Table 3, the average success rates for existing learning-from-demonstrations algorithms implemented with 3D deep learning backbones are low on both training environments and test environments. This suggests that existing algorithms might be insufficient yet to achieve high training success rates over diverse object geometries provided in our benchmark and generalize to novel objects, and our current tasks are already challenging for the community. We will work on more challenging tasks in the future, such as tasks with more complex dual-arm collaboration, and tasks involving both high-level planning and low-level physical manipulation.

---

### Official Review · Reviewer_r62r · 2021-09-19
**Enjoyable Read, Very Minor Clarifications Needed**

**Rating:** 8
**Confidence:** 3

**Strengths:**

Significance:
* As mentioned in Section 1, ManiSkill addresses shortcomings of existing datasets by testing for object-level generalization; variation within those objects, and using egocentric 3D vision (state/pointcloud/rgbd), as well as many different tracks the benchmark can be evaluated on.

Relevance:
* Numerous methods to train policies (behavioral cloning/RL with sparse/dense rewards) make this accessible for researchers from many backgrounds.

Accessibility and Accountability:
* The paper is very easy to read.  There is a discussion of the as well as a disclaimer that this is not a generic learning-from-demonstrations benchmark (line 114) which helps focus the work.
* The paper also explains how ManiSkill is limited and provides future directions to remedy this as well as an assurance that the benchmark will be continually updated and monitored.  This is very good to read!

**Weaknesses:**

There are no outstanding weaknesses -- any issues I have with the submission are minor.

Demonstrations generated by reinforcement learning may be easier to learn from and may have lower diversity than those provided by humans.  Some care should be taken when explaining why demonstrations from RL are suitable.  Moreover, discussions over the number of demonstrations and whether this is appropriate for an eventual human use-case should also be provided.

The authors could have perhaps had more discussions on why they chose the architectures they benchmarked to assure future practitioners that these are the best architectures that their research should build on.

**Additional Feedback:**

The paper was an enjoyable read overall and the issues I've raised are only minor in comparison.

**Clarity:**

The paper is very well-written.  There was a good use of bold font and underlining throughout which drew to the structure of the piece and enabled the reader to quickly find key points.  In particular, the abstract is excellently written.  Researchers who have previously worked with this benchmark who need a reminder could very quickly glean high-level information to refresh their memory.

There are some minor grammatical errors (eg. line 1, poses -> pose).

Minor point, terms such as revolute joint, prismatic joint introduced in lines 177 and 182 are introduced, and perhaps could be supported by explaining how they look like with aid of a figure.  Perhaps this could be a modification of Figure 3.

I didn't quite follow the divide-and-conquer approach mentioned in lines 108-111 or how a reward function was automatically generated when it was introduced for the first time in lines 108-111 and presumed it was expanded on in Section 2.4?  If so, a link to Section 2.4 should have been provided for better clarity.

**Correctness:**

If the demonstrations are obtained from a trained RL agent, more explanation is needed for why this trained RL agent does not suffice to solve ManiSkill.

**Documentation:**

I was able to find the GitHub repository as well as the website for an online leaderboard mentioned in the paper.
* The repository, like the paper, was well documented with clear installation instructions and example usage.  Minor point, the list of "What Can I Do with ManiSkill" seemed incomplete.
* I liked the idea of an online leaderboard as it gives those working on ManiSkill a centralised location to see the latest progress.

**Ethics:**

There are no outstanding ethical concerns presented by ManiSkill apart from those existing in RL, IL, robotics and agency.

**Relation To Prior Work:**

Much related work was referenced throughout, with up to ten papers cited in one go to support a point.  Section 1 comprehensively introduces related work (eg lines 54/55) and explains how they are lacking.  This has the added bonus of being laid out in a well-structured format.

**Summary And Contributions:**

ManiSkill is a large-scale open-source benchmark for physical manipulation skill learning over a diverse set of articulated objects from 3D visual inputs.  It consists of four tasks filtered by difficulty; the need for dual-arm collaboration and the number of DoF involved in solving a task.  Baselines are agents which use PointNets and an optional Transformer layer and are trained by a mixture of Behavioural Cloning and (Offline) RL.  Results measured in success rate show that this is a challenging benchmark.

---

> ### Author Response · Authors · 2021-09-26
> **Thanks for your encouraging feedback and suggestions!**
>
> We would like to sincerely thank you for reviewing our paper and providing constructive comments!
>
> Here we address your comments below:
>
> > Demonstrations generated by reinforcement learning may be easier to learn from and may have lower diversity than those provided by humans ...
>
> Comprehensively comparing our demonstrations with human demonstrations in terms of diversity is tricky. In addition, our benchmark is not a generic learning-from-demonstrations benchmark, so comparing our RL-generated demonstrations with human demonstrations is not the focus of our work (we focus on solving the tasks themselves). However, we do empirically observe that our RL-generated demonstrations are diverse, especially given that a large number of demonstrations have been provided.
>
> For example, in OpenCabinetDrawer, the demonstration trajectories show two different behaviors: opening the drawer by pulling the handle, or by pulling the side of the drawer out. One reason for such diverse behaviors could come from our reward design: While the multi-stage reward template we designed encourages an agent to grasp the handle of a drawer, we do not explicitly restrict how it opens the drawer. As long as the opening method is feasible and achieves the desired goal state, the trajectory will be considered successful. We included this clarification in Appendix C.2.
>
>
> > More discussions on why they chose the architectures they benchmarked
>
> PointNet is benchmarked because it is a very popular architecture for point cloud-based 3D deep learning. PointNet + Transformer is benchmarked because we believe self-attention can capture the relation between different objects and possibly provides better performance. We believe that these backbones provide great starting points for many researchers. We have revised Section 3 to address this.
>
> > More explanation is needed for why this trained RL agent does not suffice to solve ManiSkill.
>
> 1. Our RL agents used to generate the demonstrations are trained under `state` mode, which means their input includes *object states*, i.e. ground truth information about the objects. However, the evaluation protocol of ManiSkill is designed on point cloud / RGB-D visual inputs, as *object states* are not available in realistic setups and have to be estimated based on visual information.
> 2. Our demonstrations are collected by training a population of agents, each solving a single specific environment. Therefore, these RL agents are not able to generalize to unseen environments. Moreover, in Table 5 of Appendix C.2, we show that jointly training a single RL agent on a large number of environments to collect demonstrations is infeasible.

---

> > ### Comment · Reviewer_r62r · 2021-09-29
> > **Thanks for the clarifications!**
> >
> > I thank the authors for the clarifications of the points I have raised.  I believe that their inclusion would improve the paper.  As my initial score is high, I do not feel that I need to change my score.

---

### Official Review · Reviewer_5hNW · 2021-09-20
**A good benchmark for generalizable manipulation research**

**Rating:** 7
**Confidence:** 3
**Correctness:** Yes.
**Clarity:** Yes, the paper is well written and ea…

**Strengths:**

- The proposed task is challenging with the high value of the real-world application. The benchmark supports 3D perception tasks, learning of policies from interactions as well as learning-from-demonstrations, which will attract a broad set of researchers.
- The provided dataset is of high quality and the demonstrations dataset is of large scale.
- The selected baselines are reasonable and also show the challenges of the object-level generalization task, which can guide future research.

**Weaknesses:**

- The number of intra-class instances and categories is limited, which limits the generalizability to the real world.
- All the settings are in a simulation, the selected objects are hand-made CAD models. It's helpful to add some reconstructed objects which are closer to the real world and also, provide some real frames and tracks for the testing in the real world scenario to evaluate the sim-to-real gap.

**Additional Feedback:**

None

**Documentation:**

Yes, it's open-sourced with good documentation.

**Relation To Prior Work:**

Yes.

**Summary And Contributions:**

The paper introduces a generalizable manipulation skill benchmark with large intra-class topological and geometric variations. Compared with previous work, the paper focuses more on the object-level generalizability of manipulation skills. To support the new task, high-quality object data and large-scale demonstrations datasets are collected. The new challenging task may attract a wide range of research fields including 3D computer vision, reinforcement learning as well as robotics.

---

> ### Author Response · Authors · 2021-09-26
> **Thanks for your encouraging feedback and suggestions!**
>
> We would like to sincerely thank you for reviewing our paper and providing constructive comments!
>
> Here we address your comments below:
>
> > Number of intra-class instances and categories is limited
>
> The object dataset construction is an ongoing effort. We are continuously processing and modeling new objects for each existing category, and we also plan to increase the number of categories of tasks in the next version of the ManiSkill benchmark.
>
>
> > All the settings are in a simulation
>
> We agree that real-world transferability is important for a simulated manipulation benchmark. We are actively working on sim-to-real experiments and we plan to include real-world experiments and evaluations in the next version of the ManiSkill benchmark.

---

### Decision · Program_Chairs · 2021-10-10

**Decision:**

Accept

**Comment:**

This submission proposes a manipulation benchmark, with the focus of dealing with different objects in 3D simulation. All of the reviewers have rated the submission very highly, and agreed that it will be useful for a broad range of disciplines (e.g. robotics, RL). The paper is well written, the dataset is well documented, and benchmark results are provided. For this reason I recommend acceptance of this submission.